# The Yin and Yang of Antibodies in Viral Infectious Diseases

**DOI:** 10.3390/diseases13100341

**Published:** 2025-10-15

**Authors:** Jianning He, Yiu-Wing Kam, Fok-Moon Lum

**Affiliations:** 1Division of Natural and Applied Science, Duke Kunshan University, No. 8 Duke Avenue, Kunshan 215316, China; jianning.he@emory.edu (J.H.); yiuwing.kam@dukekunshan.edu.cn (Y.-W.K.); 2Rollins School of Public Health, Emory University, 1518 Clifton Road NE, Atlanta, GA 30322, USA; 3A*STAR Infectious Diseases Labs (A*IDL), Agency for Science, Technology and Research (A*STAR), Singapore 138648, Singapore; 4Lee Kong Chian School of Medicine, Nanyang Technological University, Singapore 308232, Singapore

**Keywords:** antibody, infectious diseases, humoral immunity, antibody effector functions, vaccine-induced immunity

## Abstract

Antibodies are a cornerstone of the adaptive immune response, serving as key defenders against viral infections; however, they can also act as a double-edged sword, contributing to immune-mediated pathologies. This review advances a “Yin-Yang” framework to integrate the dual activities of antibodies. The protective ‘Yin’ functions are driven by high-affinity antibodies generated through processes like somatic hypermutation and class-switch recombination. These antibodies execute viral neutralization, activate the complement system, and engage Fc receptors (FcRs) to drive antibody-dependent cellular cytotoxicity (ADCC) and phagocytosis. These mechanisms form the immunological basis of effective vaccines, which aim to elicit durable and functionally specialized antibody isotypes like IgG and mucosal IgA. Conversely, the pathogenic ‘Yang’ of the response can be detrimental. This includes antibody-dependent enhancement (ADE) of infection, notably observed with flaviviruses, and the development of autoimmunity through mechanisms like molecular mimicry and bystander activation, which can lead to conditions such as multiple sclerosis and Guillain-Barré Syndrome. The balance between protection and pathology is tipped by a confluence of factors. These include viral evasion strategies like antigenic mutation and glycan shielding, as well as host-based determinants such as genetic polymorphisms in FcRs, immune history, and the gut microbiome. Understanding these molecular determinants informs the rational design of next-generation interventions. Promising strategies, such as Fc-region glyco-engineering and the design of tolerogenic vaccines, aim to selectively promote protective functions while minimizing pathological risks, offering a clear path forward in combating viral threats.

## 1. Introduction

The humoral immune response, which is part of the adaptive immune response alongside the T-cell-mediated responses, is mediated by B cells [1]. Naive B cells express IgM or IgD on their surface, serving as receptors for antigen recognition. After immunization or infection, activated naive B cells undergo class switching recombination (CSR), which involves a change in the heavy-chain constant (C) region to another isotype. This results in a change in the B-cell surface antibodies from IgM/IgD to secretable IgG, IgE, or IgA [2]. CSR is guided by cytokines, T-cell help, antigen presentation, B-cell receptor engagement, and transcription factors. These signals determine the eventual antibody isotype, customizing the effector function of the antibody and enhancing its ability to eliminate the specific pathogen that induced the response [3,4]. Activated B cells can also undergo another process known as somatic hypermutation (SHM), which involves introducing mutations in the variable (V) region of the heavy and light chains of the immunoglobulin. This process leads to the production of antibodies with increased affinity and avidity for their target antigen [5].

The significance of antibodies is further underscored by their status as a key output of vaccination, with most vaccines relying on the generation of protective antibodies to confer immunity against infectious diseases [6]. The emphasis on eliciting robust antibody responses through vaccination highlights the critical importance of these molecules in preventing infection and disease. However, the complex relationship between antibodies and pathogens is a double-edged sword [7]. On one hand, antibodies can provide life-saving immunity against deadly diseases. On the other hand, they can also contribute to immune evasion, antibody-dependent enhancement of infection, and autoimmune disorders [7] (Figure 1). Despite these challenges, advances in antibody engineering and technology have opened up new avenues for the development of antibody-based therapies, offering hope for the prevention and treatment of emerging infectious diseases [8].

Despite extensive research, critical gaps remain in our understanding of the precise molecular and cellular switches that dictate whether an antibody response is protective or pathogenic. For instance, how do subtle variations in antibody glycosylation or FcγR genetics tip the balance during a secondary flavivirus infection? Furthermore, how can vaccine design be optimized to selectively induce protective functions while minimizing the risk of ADE or autoimmunity? The primary objective of this review is to address these questions by synthesizing evidence from diverse viral systems under a unifying ‘Yin-Yang’ framework. We will explore the molecular determinants of antibody function and discuss how this knowledge can be leveraged for the rational design of safer and more effective immunotherapies and vaccines.

## 2. Materials and Methods

This manuscript is a narrative and descriptive literature review designed to synthesize the dualistic ‘Yin-Yang’ roles of antibodies in viral infectious diseases. A structured, conceptually driven literature search was conducted across PubMed, Scopus, and Google Scholar using keywords related to antibody functions. The search prioritized publications from January 2015 to present to ensure currency while also including seminal foundational articles for historical context. Selection was limited to peer-reviewed original research and authoritative reviews in English, excluding articles with poorly described methodologies. The selected evidence was then thematically synthesized, organizing findings to first establish the protective ‘Yin’ mechanisms (e.g., neutralization, ADCC) and subsequently explore the detrimental ‘Yang’ roles (e.g., ADE, autoimmunity), thereby constructing a coherent narrative that connects fundamental immunology to clinical implications and the design of future interventions.

## 3. Role of Antibodies in Viral Infections

Antibodies play an important role in the immune response against viral infections. Their functions extend beyond simple neutralization, encompassing mechanisms such as complement activation, phagocytosis, and antibody-dependent cellular cytotoxicity (ADCC). Depending on the pathogen, some antibodies confer long-term immunity, while others wane over time, affecting reinfection susceptibility and vaccine efficacy [9]. While antibodies are primarily protective, under certain circumstances, they contribute to disease pathology. These pathogenic effects arise due to unintended immune activation, molecular mimicry, or immune complex formation, leading to complications such as autoimmune diseases, antibody-dependent enhancement (ADE) of infection, and ineffective immune responses due to antigenic variation. Understanding these mechanisms will be crucial for mitigating the risks associated with antibody-based therapies and vaccines, and aid in the future development of safer and more effective treatments.

### 3.1. Protective Role of Antibodies in Viral Infections

#### 3.1.1. Neutralization

Neutralization is one of the primary mechanisms by which antibodies confer protection against pathogens. The process of neutralization involves antibodies binding to viruses, blocking their ability to attach to host cells [10]. This type of neutralization is termed “steric hindrance”, where the physical presence of the antibody blocks viral entry points [10,11]. Another significant mechanism is through either the induction or blocking of conformational changes on the virus, rendering the virus non-infectious [11,12]. These conformational changes can either directly block the virus’s ability to interact with host cell receptors or expose the virus to other components of the immune system, such as complement proteins, which can further inactivate the virus [11]. Antibodies can also facilitate the aggregation of viruses, making it more difficult for the viruses to navigate to and infect host cells [10,13]. This aggregation also makes it easier for immune cells to recognize and eliminate the virus. In this process, antibodies act as opsonins, which mark the viruses for downstream destruction [14]. In addition, antibodies can also block endosomal cleavage or endosomal receptor binding. This is critical for viruses that enter endosomes. In an indirect context of neutralization, antibodies can block viral egress, leading to the accumulation of virus progenies at the surface of the infected cells [12]. Robust neutralization of respiratory syncytial virus (RSV) has been achieved with site-IV–targeting F-protein monoclonal antibodies such as 5B11 [15]. In a phase 1b/2a trial, a single prophylactic dose reduced RSV viral load by >95% and cut hospitalizations in infants by ~70% compared with placebo, showing that high-affinity F-protein IgG can translate into clinically meaningful protection [15].

#### 3.1.2. Tailoring the Response: Class-Switch Recombination and Fc-Mediated Effector Functions

While high affinity is essential for antigen binding, the functional capacity of an antibody to eliminate a viral threat is determined by its isotype, which is dictated by the constant (C) region of its heavy chain [16]. Within the germinal center, B cells undergo a second crucial DNA modification process known as class-switch recombination (CSR). CSR replaces the default Cμ gene (encoding IgM) with a downstream Cγ, Cα, or Cε gene, leading to the production of IgG, IgA, or IgE isotypes, respectively [17]. This process is precisely directed by the cytokine milieu established by Tfh cells and other local immune cells in response to the specific viral challenge. For instance, the cytokine interferon-gamma (IFN-γ) promotes switching to IgG subclasses (like IgG1, IgG2a and IgG2c in mice), which are potent inducers of cell-mediated cytotoxicity, while transforming growth factor-beta (TGF-β) is critical for switching to IgA, the primary isotype for mucosal immunity [18,19].

The structural differences in the Fc region of each antibody isotype allow the immune system to tailor its effector response. These Fc regions engage with a diverse family of FcRs expressed on various immune cells, thereby bridging the humoral and cellular arms of immunity. For example, IgG antibodies, like IgG1 and IgG3, are powerful mediators of ADCC, where they coat virus-infected cells and engage Fcγ receptors on NK cells, triggering the release of cytotoxic granules [20]. Furthermore, IgG and IgM can activate the classical complement pathway, leading to the formation of a membrane attack complex that can lyse enveloped viruses or infected cells [21]. IgA, which is secreted across mucosal surfaces, acts as a first line of defense by neutralizing viruses at entry portals, a process known as immune exclusion [22]. The strategic deployment of different antibody isotypes, each with a distinct Fc-mediated functional profile, is therefore indispensable for a comprehensive and effective antiviral defense.

#### 3.1.3. Activation of the Complement System

The complement system is an integral component of the innate immune response, consisting of over 50 proteins that interact in a highly regulated manner to combat infections [23]. Activation of this system leads to a cascade of events that initiates key immune processes such as the recruitment of inflammatory cells, opsonization of pathogens, and formation of the membrane attack complex [23]. These processes enhance pathogen clearance and bridge innate and adaptive immunity [23]. While complements are generally protective against infectious agents, it is worthwhile to note that complements have also been implicated in the modulation of ADE of flavivirus infections [24]. A comprehensive review of this phenomenon was previously published by Byrne and Talarico [24].

#### 3.1.4. Antibody-Mediated Phagocytosis

Another protective mechanism of antibodies is through antibody-mediated phagocytosis, in which antibodies act as opsonins to enhance the recognition and uptake of pathogens by immune cells such as macrophages and neutrophils [14]. This process is initiated when antibodies interact with the Fc receptors (FcRs) expressed on the surface of the immune cells. This interaction triggers downstream signaling pathways that promote the engulfment of antibody-coated particles. Among the key FcRs, Fcγ receptors (FcγRs) are known to mediate phagocytosis. FcγRs are classified into several classes in humans, including FcγRI (CD64), FcγRIIa (CD32a), FcγRIIb (CD32b), FcγRIIc (CD32c) FcγRIIIa (CD16a) and FcγRIIIb (CD16b), which differ in their affinity for IgG and their ability to mediate inflammatory or anti-inflammatory responses. Except for FcγRIIb and FcγRIIIb, all other FcγRs contain intracellular immunoreceptor tyrosine-based activating motifs (ITAM). Instead, FcγRIIb contains the immunoreceptor tyrosine-based inhibitory motifs (ITIM) and FcγRIIIb is a decoy receptor and lacks intracellular signaling motifs [25,26].

Interestingly, it was reported recently that other host factors could affect FcγRs function. In an animal model of rheumatoid arthritis, it was determined that Dectin-1, a C-type lectin receptor, enhances the binding of IgG to the low-affinity FCγRIIb. This interaction reprograms the monocytes, ultimately causing an inhibition of osteoclastogenesis [27]. Other factors, such as glycosylation, could also impact the functions of antibodies and their binding to the FcRs [28]. The role of glycosylation of antibodies in the context of infectious diseases [29] and vaccine designs [30] has been comprehensively reviewed.

#### 3.1.5. Antibody-Dependent Cellular Cytotoxicity (ADCC)

Antibodies can also participate in the elimination of virus-infected cells through a mechanism known as antibody-dependent cellular cytotoxicity (ADCC), which leverages the immune system’s effector cells. During viral infections, antibodies bind to viral antigens on the surface of infected cells and immune cells, such as natural killer (NK) cells, macrophages, neutrophils, and eosinophils, which recognize these antibody-coated targets through their surface FcRs. This triggers a response that ultimately leads to the infected cell’s death [31]. Specifically, NK cells express FcγRIIIa that binds to the Fc region of IgG antibodies, inducing the release of cytotoxic granules containing perforin and granzymes [32]. This mechanism is crucial in controlling viral infections, including influenza, which is notorious for its high mutation rate in the viral glycoprotein, hemagglutinin (HA) [33]. Notably, it was reported that ADCC-mediating antibodies targeting a specific 14 amino acid fusion peptide sequence at the N-terminus of the HA2 subunit are able to induce ADCC against a wide range of influenza viruses [34,35]. Interestingly, the binding affinity between the antibodies and FcγRIII on NK cells determines the strength of ADCC. Recently, it was reported that afucosylation of broadly neutralizing antibodies targeting human immunodeficiency virus (HIV)-1 envelope glycoprotein potentiates activation and degranulation of NK cells, marked by the increase in CD107^+^IFNγ^+^ cells. It was also reported that afucosylated antibodies could overcome the inhibitory signals in exhausted PD-1^+^ and TIGIT^+^ NK cells, leading to their activation, thereby amplifying ADCC and accelerating viral clearance [36]. In dengue infection, afucosylation of IgG is associated with severe dengue disease and has been associated with thrombocytopenia leading to significant loss of platelets [37]. Nevertheless, in addition to fucose, IgG antibodies can also be modified via the addition of other glycan moieties such as galactose and sialic acid [38]. However, the effects of both galactosylation and sialylation on IgG functions remain debatable, as contradictory effects have been reported [29], exemplifying the importance of gaining a greater understanding of glycosylation on antibody function in human diseases [39].

### 3.2. Detrimental and Pathological Roles of Antibodies in Viral Infections

#### 3.2.1. Autoimmunity and Autoantibodies

Autoimmunity can arise when viral infections induce immune responses that mistakenly target host tissues through mechanisms such as dysregulation of FcγR-mediated pathways, autoantibody production, molecular mimicry, bystander activation, and epitope spreading [40].

Dysregulation of FcγR-mediated pathways has been linked to autoimmune diseases such as systemic lupus erythematosus (SLE), rheumatoid arthritis (RA), and anti-neutrophil cytoplasmic antibody (ANCA)-associated vasculitis. In these conditions, autoantibodies mediate pathology through distinct mechanisms. They form pathogenic immune complexes in SLE [41], engage activating FcγRs to drive joint inflammation in RA [42], and activate neutrophils to cause vascular injury in ANCA-associated vasculitis [43].

Autoantibodies generated during infections can be transient, resolving after clearance of the pathogen, or persist long-term, leading to chronic autoimmune conditions [40]. For example, autoantibodies against interferon-gamma (IFN-γ) generated during SARS-CoV-2 infection can persist, exacerbating long COVID-19 and contributing to severe acute respiratory syndrome (SARS) by impairing host immune responses [44]. Clinical studies indicate that the presence and higher titers of anti-IFN-γ autoantibodies correlate significantly with severe or critical COVID-19 cases, suggesting their potential role as biomarkers for predicting disease severity [45]. These autoantibodies can functionally neutralize IFN-γ by effectively inhibiting its signaling pathways, which phosphorylate STAT1, thereby impairing critical antiviral defenses and exacerbating disease outcomes [45].

Molecular mimicry, which depends on structural similarity between viral antigens and host proteins, is mediated by cross-reactive antibodies produced by the B cells during the infection. A notable example is Epstein–Barr Virus (EBV) (Table 1), where antibodies generated against EBV nuclear antigen 1 (EBNA1) cross-react with similar epitopes found on host myelin proteins. This cross-reactivity results in autoreactive B cell activation and production of pathogenic autoantibodies, contributing to autoimmune pathology such as multiple sclerosis (MS) [46]. In fact, molecular mimicry between EBNA1 and other CNS proteins such as anoctamin-2 (ANO2) [47], alpha-B crystallin (CRYAB) [48] and myelin basic protein (MBP) [49] has also been described.

It was also recently reported that children with multisystem inflammatory syndrome (MIS-C), develop a unique immune response following SARS-CoV-2 infection, targeting a distinct domain within the viral nucleocapsid protein that bears sequence similarity to the self-protein SNX-8 [50]. Likewise, arboviruses like Zika virus (ZIKV) and dengue virus (DENV) have also been implicated in autoimmunity through the production of autoantibodies and molecular mimicry [51]. For instance, ZIKV neutralizing antibodies have been shown to cross-react with neuronal membrane gangliosides in ZIKV-associated Guillain-Barré Syndrome (GBS) cases, suggesting that molecular mimicry is a potential mechanism used by ZIKV to cause neurological damage [52]. In these GBS patients, autoantibodies against host glycolipids were detected [52]. Similarly, in DENV, it was reviewed by Zhou et al. (2025) that molecular mimicry could lead to the production of autoantibodies targeting platelets, endothelial cells and coagulatory factors, ultimately affecting thrombocytopenia and plasma leakage during severe dengue [51].

Bystander activation describes the non-specific stimulation of immune cells during infection, which can inadvertently activate autoreactive B and T cells. Chronic cytomegalovirus (CMV) infection exemplifies this mechanism by persistent immune activation, potentially leading to autoimmune diseases such as systemic lupus erythematosus (SLE) and rheumatoid arthritis (RA) [53]. In COVID-19 patients, bystander activation of polyclonal autoreactive B cells has been reported. Activation leads to the production of a broad range of autoantibodies, albeit none of the elevated autoantibodies were associated with disease severity [54]. Likewise, infection with either SARS-CoV-2 or influenza, led to an expansion of CCR6^+^CXCR3^−^ bystander memory B cells (MBCs) alongside the CCR6^+^CXCR3^+^ virus-specific MBCs in the lungs of the infected animals. These bystander MBCs differ in their origin and transcriptional programs and elicit antibodies that are non-specific and offer no protection [55]. The role of such bystander cells in the context of infection should be further studied.

Epitope spreading refers to the progressive diversification of immune responses, initially targeting a limited antigenic site and subsequently extending to other epitopes within the same or different antigens. For example, in hepatitis C virus (HCV) infection, initial immune responses may broaden over time, contributing to autoimmune manifestations like cryoglobulinemia and Sjögren’s syndrome [56]. Interestingly, in the context of EBV infection, Sattarnezhad and team hypothesized intramolecular epitope spreading as a mechanism to increase the breadth of antibody responses against glial cell adhesion molecule (GlialCAM), potentially increasing its potential to cause harm [57]. In mice, injections with EBNA1 peptides of different length elicited cross-reactive antibodies to dsDNA, different from the cross-reactive epitope [57,58].

To balance antibody protection with safety, we regard the reduction in antibody-mediated autoimmunity as a core design objective rather than an afterthought. We recommend three complementary measures: (i) Fc-region glyco-engineering and judicious IgG subclass selection to bias Fcγ-receptor engagement toward inhibitory or anti-inflammatory pathways [59]; (ii) tolerogenic adjuvants and epitope choices that minimize molecular mimicry of self-antigens [60]; and (iii) proactive immunomonitoring to detect early signs of autoreactivity in at-risk recipients [61]. Integrating these safeguards will help ensure that next-generation vaccines and therapeutic antibodies deliver maximal benefit while maintaining an acceptably low autoimmune liability.

**Table 1 diseases-13-00341-t001:** Examples of Virus-Induced Autoantibodies and Associated Autoimmune Diseases. This table summarizes select autoimmune diseases and conditions linked to viral infections. For each condition, it lists the associated viruses, the specific autoantibodies produced or the self-antigens they target, and the proposed immunopathogenic mechanisms responsible for the development of autoimmunity.

Autoimmune Disease/Condition	Associated Virus(es)	Autoantibodies/Autoantigen Targets	Proposed Immunopathogenic Mechanism(s)
Systemic Lupus Erythematosus (SLE)	Epstein–Barr Virus (EBV) [62]	Anti-nuclear antibodies (ANA), anti-dsDNA, anti-Sm, anti-CL/beta2-GPI complex, anti-RNP, anti-Ro/SSA, anti-La/SSB, antiphospholipid antibodies (aPL) [63]	FcγR dysregulation [64]
Rheumatoid Arthritis (RA)	Epstein–Barr Virus (EBV) [65], Cytomegalovirus (CMV) [66]	Rheumatoid Factor (RF) [42], anti-citrullinated protein antibodies (ACPAs) [67]	Bystander activation [68]
ANCA-Associated Vasculitis	Epstein–Barr Virus (EBV) [69], Hepatitis B Virus (HBV) [70]	Anti-neutrophil cytoplasmic antibodies (ANCAs) [43]	Neutrophil activation [71]
Multiple Sclerosis (MS)	Epstein–Barr Virus (EBV) [72]	Oligoclonal IgG bands (CSF); no single disease-defining serum autoantibody [73]	Molecular mimicry; Epitope spreading [74]
Guillain-Barré Syndrome (GBS)	Epstein–Barr Virus (EBV) [75], Cytomegalovirus (CMV) [76], Influenza [77], Zika Virus (ZIKV) [52]	Anti-ganglioside antibodies [52]	Molecular mimicry [51]
Multisystem Inflammatory Syndrome in Children (MIS-C)	SARS-CoV-2 [50]	Antibodies targeting viral nucleocapsid cross-reactive with self-protein SNX-8 [50]	Molecular mimicry [50]
Thrombocytopenia/Plasma Leakage	Dengue Virus (DENV) [37]	NS1 mimicry, endothelial and platelet autoimmunity [78]	Molecular mimicry [78]
Sjögren’s Syndrome	Epstein–Barr Virus (EBV) [79], Hepatitis C Virus (HCV) [56]	Anti-Ro/SSA (Ro52, Ro60), Anti-La/SSB, RF, ANA [80]	Epitope spreading [81]

#### 3.2.2. Therapeutic Strategies to Mitigate Antibody-Mediated Pathologies

Recent advancements in protein and vaccine engineering have paved the way for innovative strategies aimed at reducing the pathogenic potential of antibodies. One promising approach is Fc-region glyco-engineering, which involves modifying the glycan structures on the antibody’s Fc domain. By altering these glycans, it is possible to shift the antibody’s binding preference from activating FcγRs to the inhibitory receptor FcγRIIb. This shift effectively dampens inflammatory responses, transforming a potentially pathogenic antibody into a therapeutic agent [82]. For instance, increasing the sialylation of the Fc region has been shown to enhance affinity for FcγRIIb, thereby promoting anti-inflammatory activity [83]. This strategy is being explored for the treatment of autoimmune and inflammatory diseases where antibody-mediated pathology is a key driver.

In parallel with modifying existing antibodies, significant efforts are being directed towards the rational design of tolerogenic vaccines. These vaccines are engineered to avoid epitopes with high sequence or structural homology to self-antigens, which is a crucial step in minimizing the risk of inducing cross-reactive autoantibodies through molecular mimicry [84]. Furthermore, the development of engineered therapeutic antibodies has provided a powerful tool for targeted intervention in autoimmune conditions. These monoclonal antibodies can be designed to specifically deplete autoreactive B cells. An example is Rituximab, an anti-CD20 antibody that effectively removes B cells and is used to treat various autoimmune diseases [85]. Other engineered antibodies are designed to block key inflammatory pathways, such as those targeting cytokines like TNF-α or IL-6, offering a more direct approach to controlling inflammation [86]. These strategies represent the forefront of efforts to control antibody-mediated pathologies.

#### 3.2.3. Antibody-Dependent Enhancement (ADE)

ADE occurs when non-neutralizing or sub-neutralizing antibodies facilitate viral attachment (extrinsic ADE) or entry (intrinsic ADE) into host cells, paradoxically worsening the infection [87,88]. This phenomenon has been extensively studied in the context of Flavivirus infections, such as DENV. Antibodies generated against one serotype of DENV can bind but fail to neutralize a different serotype, forming immune complexes that interact with FcγRs on monocytes, macrophages, or dendritic cells. This process enhances viral uptake and replication, leading to severe disease manifestations like dengue hemorrhagic fever and dengue shock syndrome [89]. Clinically, this mechanism can significantly increase disease severity and contribute to severe complications such as myocarditis, characterized by elevated cardiac enzymes, arrhythmias, heart failure, and increased mortality risk [90]. In a clinical setting, patients experiencing antibody-dependent enhancement of dengue infection demonstrated notably higher levels of inflammatory markers like C-reactive protein (CRP), prolonged prothrombin time (PT), and activated partial thromboplastin time (aPTT), as well as an increased requirement for intensive care unit admission and longer hospital stays [90]. Furthermore, related viruses can induce cross-reactive antibodies that may impact the outcome of infection caused by other viruses within the same viral family. For example, it was reported that the presence of Japanese encephalitis virus (JEV) neutralizing antibodies was associated with an increased presence of DENV symptomatic infection compared to asymptomatic infection, with the symptomatic infection also lasting for a longer duration [91]. Likewise, in a separate study, vaccination with a single dose of inactivated Vero cell-derived JEV vaccine led to production of antibodies capable of enhancing DENV infection at sub-neutralizing levels [92]. Interestingly, pre-existing immunity elicited by the JEV SA14-14-2 vaccine conferred in vivo protection against Zika virus in a mouse model [93]. However, the presence of antibodies to both DENV and West Nile virus (WNV) was found to enhance ZIKV infection [94,95]. These examples show that ADE is a complex and dynamic phenomenon that needs to be better understood, particularly in the context of viral infection and vaccine development.

Nevertheless, ADE has also been observed in other viral infections, including those caused by coronaviruses. During the COVID-19 pandemic, some studies suggested that antibodies generated against the SARS-CoV-2 spike protein might enhance viral entry under certain conditions. These effects are mediated through FcγRIIa and FcγRIIIa expressed on immune cells, which bind to antibody-virus complexes and facilitate endocytosis [96]. While evidence of ADE in SARS-CoV-2 infection remains limited and controversial, it raises concerns about the design of vaccines and monoclonal antibody therapies, particularly in populations previously exposed to the virus or related coronaviruses [23,96].

Beyond flaviviruses and coronaviruses, ADE has been demonstrated for other viruses such as respiratory syncytial virus (RSV) [97], measles virus [98], Ebola virus [99], and alphaviruses [100,101]. The underlying mechanisms of ADE differ among viruses, but they generally involve FcR-mediated viral entry and increased inflammatory responses leading to tissue damage [102]. Understanding the molecular determinants of ADE is essential to designing safer vaccines and therapeutic antibodies that avoid this adverse immune phenomenon. Moreover, vaccine designs should focus on eliciting robust neutralizing antibody responses while reducing non-neutralizing antibodies. Similarly, antibody concentrations can also dictate the functional outcome of antibodies. When present in inadequate levels, neutralizing antibodies could shift their role from protection to enhancement of infection severity via ADE. This underscores the need for vaccines to induce and maintain high-titer neutralizing antibodies. Other factors to consider include the careful selection of antigens, coupled with the optimization of vaccine formulations and consideration of immune responses in diverse populations, including those with pre-existing immunity to related viruses, which will be crucial in the development of safer and more effective vaccines against viruses where ADE is a concern [103]. This is especially important in regions with a high prevalence of viral infections and limited access to healthcare resources.

## 4. Non-Neutralizing Antibodies (NNAbs)

Non-neutralizing antibodies (NNAbs) constitute an immunologically versatile subset whose activities can tip toward either host protection or pathology. When properly regulated, NNAbs engage FcγRs and complement to drive ADCC, phagocytosis, and other Fc-effector mechanisms that enhance the neutralizing response and broaden protection. Conversely, the same Fc-mediated interactions can facilitate ADE of infection, promote immune-complex deposition, or fuel autoreactive inflammation, thereby worsening disease.

A significant function of NNAbs is antibody-dependent cellular phagocytosis (ADCP). This mechanism involves the opsonization of pathogens or infected cells by NNAbs, effectively marking them for engulfment by phagocytic cells such as macrophages and neutrophils [104]. The Fc regions of NNAbs bind to FcRs on phagocytes, enabling the internalization and degradation of immune complexes. ADCP has been identified as a crucial protective mechanism in various infections, including HIV, where NNAbs help control viral replication and restrict the spread of the virus [105].

Additionally, NNAbs play crucial roles in vaccine-induced immunity. The RV144 HIV vaccine trial identified NNAbs as correlates of protection, highlighting their ability to reduce viral loads and transmission via FcR-mediated functions. Such findings emphasize the importance of eliciting NNAbs through vaccine strategies, complementing neutralizing antibody responses [105].

Beyond these effector functions, NNAbs are involved in modulating immune responses by enhancing antigen presentation by dendritic cells. Immune complexes formed by NNAbs bind to Fc receptors (FcRs) on dendritic cells, facilitating the uptake, processing, and subsequent presentation of antigens to T cells [104]. This interaction not only promotes dendritic cell maturation but also enhances adaptive immune responses, bridging innate and adaptive immunity. Notably, glycan-reactive monoclonal antibodies 2G12, PTG126, and PTG128 that arise during HIV-1 infection recognize the S protein of SARS-CoV-2 yet remain non-neutralizing, and have not been linked to ADE or adverse COVID-19 outcomes [106]. However, 2G12 can bind to analogous oligomannose motifs on influenza A hemagglutinin and can neutralize the virus, showing that glycan-focused cross-reactivity may confer heterologous protection [106]. Such observations emphasize the role of NNAbs in infectious disease outcomes and vaccine strategy design.

NNAbs can also mediate detrimental effects through ADE. At sub-neutralizing levels, NNAbs can bind to viral antigens without effectively preventing viral entry. This binding forms virus-antibody complexes that interact with FcγRs on immune cells, including monocytes, macrophages, or dendritic cells. These interactions facilitate increased internalization of viruses, enhancing viral replication within these cells [23]. Furthermore, NNAbs can trigger complement activation, leading to inflammation and tissue damage. Such mechanisms have been notably observed in viral infections like dengue virus, highlighting a critical risk factor in vaccine development [23].

## 5. Factors Influencing Antibody-Mediated Immunity

Pathogens employ various immune evasion strategies to circumvent detection and elimination by the host’s immune system. These mechanisms ensure their survival, persistence, and ability to establish long-term infections.

### 5.1. Viral-Based Factors

Pathogens employ various strategies to evade the host immune system. Pathogens like influenza and SARS-CoV-2 undergo frequent mutations in their surface proteins, allowing them to evade antibody recognition. For example, mutations in the Spike protein of SARS-CoV-2 enable the viruses to escape neutralizing antibodies [23]. This mechanism is also prominent in the influenza virus, which undergoes antigenic drift and shift in its hemagglutinin and neuraminidase proteins [107]. Similarly, human immunodeficiency virus (HIV) continuously mutates its envelope glycoprotein (gp120), creating a moving target for the immune system and thwarting the development of effective neutralizing antibodies [108]. This may lead to prolonged or severe disease and increases the transmission of these viruses. As such, it is important that researchers are actively monitoring pathogen evolution, which will help in predicting the emergence of new strains and inform vaccine development.

Epstein–Barr Virus (EBV) achieves immune evasion by entering a latent state during which it minimizes its antigenic profile. In latency, EBV expresses a limited set of viral proteins, such as EBNA-1 and LMP-2, avoiding detection by host antibodies and cytotoxic T cells [109]. Another mechanism utilized by EBV employs the viral protein BGLF5 to degrade host RNA and inhibit the expression of key antiviral molecules. This reduces the efficiency of antigen presentation and neutralizing antibody production, enabling the virus to evade both innate and adaptive immunity [109].

Pathogens can also directly interfere with antibody-mediated responses. Herpes simplex virus (HSV) encodes glycoproteins such as gE and gI, which form a complex that binds to the Fc region of IgG. This Fc-binding activity prevents the interaction of IgG with FcRs on phagocytes, effectively neutralizing antibody-dependent cellular phagocytosis [110].

It is also possible for viruses to protect key epitopes from antibody recognition through glycan shielding or structural rearrangements. HIV is a prominent example, as it incorporates a dense glycan shield over its envelope protein, preventing antibody binding to conserved epitopes [111]. Additionally, structural rearrangements in the spike protein of coronaviruses, including SARS-CoV-2, conceal receptor-binding domains during specific stages of the viral life cycle, reducing antibody access [112].

### 5.2. Host-Based Factors

The duration and effectiveness of antibody responses vary among individuals due to genetic differences [113,114], immune history [115,116,117], and co-existing conditions [118]. Some individuals generate long-lasting protective antibodies, while others exhibit rapid waning, influencing susceptibility to reinfection and vaccine efficacy [118]. Factors such as HLA haplotypes, Fc receptor polymorphisms [119], and baseline inflammation levels [120] can modulate the quality and persistence of antibody responses. Interestingly, in a recent report on how immune history could affect antibody responses, Lv et al. demonstrated that in mice, antibodies raised against pre-2009 H1N1 strains can significantly affect both anti-HA and anti-NA antibody responses when these animals were exposed to the 2009 pandemic H1N1 strain [117].

The gut microbiome, which participates in immune system development and function, can affect the quality and magnitude of the host’s humoral responses to an infection [121]. Dysbiosis, which is an imbalance in the gut microbiome, compromises immune function, and this could substantially reduce the effectiveness of antibody responses. Consequently, future research should prioritize modulating the host microbiome to bolster immune function for better vaccine efficacy [122].

Non-genetic factors such as diet, lifestyle, infection history, and vaccination status can also influence antibody-mediated immunity [117,123,124,125]. For example, nutritional deficiencies in vitamin D and zinc have been linked to impaired B cell function and suboptimal vaccine responses [126]. Immune imprinting, rooted in the fundamentals of immunological memory, can impact vaccination-induced antibody responses, especially against viruses that undergo rapid evolution (e.g., SARS-CoV-2) [125]. Additionally, chronic infections involving tuberculosis or helminth infestations can skew immune responses toward a regulatory phenotype, affecting vaccine efficacy and antibody durability [127,128]. Adequate rest, effective stress management, regular physical activity, avoidance of smoking, moderation in alcohol consumption, and minimal exposure to environmental pollutants are crucial factors that further influence the immune system’s functionality. Understanding these interactions is critical for developing strategies to enhance immune resilience and improve vaccine outcomes.

## 6. Conclusions

The antibody response to viral infection is a delicate balance, a true ‘Yin-Yang’ of protection and pathology. This review has synthesized how fundamental immune processes—from germinal center dynamics to class-switch recombination—generate a functionally diverse antibody arsenal. These antibodies serve as the cornerstone of antiviral defense, neutralizing virions and orchestrating potent Fc-mediated effector functions like ADCC and complement activation, which form the basis of effective vaccination.

On the other hand, this same response can become detrimental. We have highlighted how these antibodies can drive pathologies like antibody-dependent enhancement or, through mechanisms like molecular mimicry, trigger the production of autoantibodies that lead to debilitating autoimmune conditions such as Guillain-Barré Syndrome and virus-induced Systemic Lupus Erythematosus. The final outcome is not random but is tipped toward protection or harm by a complex interplay of host genetics, immune history, and viral evasion tactics.

Harnessing this dualistic understanding is therefore crucial for the next generation of medical interventions against viral threats. Looking forward, the rational design of safer vaccines and immunotherapies depends on our ability to selectively promote protective functions while minimizing harm. Promising strategies, such as Fc-region glyco-engineering to dampen inflammatory signals and designing tolerogenic vaccines to avoid cross-reactivity with self-antigens, represent the forefront of this effort. Ultimately, by embracing the complexity of the antibody’s ‘Yin-Yang’ nature, we can develop more precise and effective tools to control viral diseases.

## Figures and Tables

**Figure 1 diseases-13-00341-f001:**
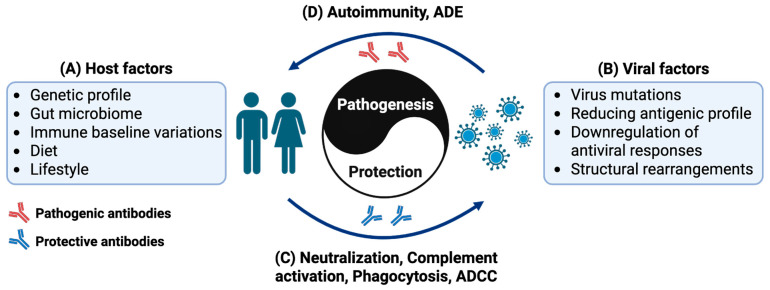
Schematic diagram illustrating the immune interaction between the host and virus, conceptualized as an immune “Yin and Yang”. It provides a visual framework for understanding the dynamic process of immune attack and defense in infectious diseases. (**A**) On the left is the human host with its immune response shaped by factors such as genetics, intestinal flora, immune baseline levels, diet, and lifestyle. (**B**) The right side represents viruses employing strategies like high-frequency mutation, antigen simplification, inhibition of antiviral molecules, and structural rearrangement to evade the host‘s immunity. Antibodies play a crucial role in the immune responses, mediating the interaction between the host and the pathogen. The outcome of this dynamic interaction depends on the antibody activities, either (**C**) host protective driving virus elimination through neutralization, complement activation, phagocytosis and antibody dependent cellular cytotoxicity (ADCC), or (**D**) virus exploitative, leading to onset of autoimmunity, or enhanced infection mediated through antibody-dependent enhancement (ADE). This concept metaphorically describes the bifunctional roles of antibodies in immune responses, where the equilibrium between the host’s immune defenses and a pathogen’s evasion strategies determines the outcomes (health or disease).

## Data Availability

No new data were created in this review article.

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
