# Peer review of "The Yin and Yang of Antibodies in Viral Infectious Diseases"

_diseases, 2025, doi:10.3390/diseases13100341_

Round 1
Reviewer 1 Report
Comments and Suggestions for Authors
In this review, the author discusses the advantages and disadvantages of antibodies, essentially outlining both their beneficial and detrimental functions. While the content is informative, the manuscript does not provide much novel insight. An important limitation is that the review does not clearly highlight future directions in this area. In addition, the advantages and disadvantages are presented separately, without sufficient integration. It would strengthen the manuscript if the author could connect these aspects. For example, antibody concentration can influence antibody-dependent enhancement (ADE), where the cutoff level may determine whether the effect is neutralization or ADE. Addressing such interconnections would improve the impact and depth of the review.
Reviewer 2 Report
Comments and Suggestions for Authors
Major Reviewer’s Comments to the Authors:
- The review article is well-written and presents the subject matter in a coherent manner. However, the scope of the topic appears to be very broad, as infectious diseases can be caused by a diverse range of pathogens, including viruses, bacteria, fungi, and parasites. Despite this wide spectrum, the manuscript predominantly emphasizes viral infections, with limited discussion on other classes of pathogens. To ensure conceptual clarity and scientific consistency, the authors may consider either expanding the discussion to comprehensively cover antibody responses across all major groups of infectious agents or, alternatively, refining the title to reflect the predominant focus on viral diseases. Such an adjustment would enhance the precision, relevance, and overall impact of the manuscript.
- Title
The proposed title for the manuscript “The Yin and Yang of Antibodies in Infectious Diseases” needs to be improved or modified, as infectious diseases can be caused by a diverse range of pathogens, including viruses, bacteria, fungi, and parasites. Despite this wide spectrum, the manuscript predominantly emphasizes viral infections, with limited discussion on other classes of pathogens. To ensure conceptual clarity and scientific consistency, the authors may consider either expanding the discussion to comprehensively cover antibody responses across all major groups of infectious agents or, alternatively, refining the title to reflect the predominant focus on viral diseases. Such an adjustment would enhance the precision, relevance, and overall impact of the manuscript.
3. Abstract
i. Line No-17-18 Protective mechanisms……………….. basis of many effective vaccines.
ii. Line No. 24 While all these factors complicate……………
iii. Several important factors appear to be discussed in the manuscript but are not reflected in the abstract. For completeness and balance, these factors should also be briefly highlighted in the abstract and vice versa.
- The abstract needs refinement by addressing the role of effective vaccines, which is a missing component in manuscript. It should briefly outline the major types of vaccines and their effectiveness in preventing infectious diseases, while also linking these to antibody-mediated immune responses for both immediate and long-term protection
- Which all other factors none is mentioned in abstract, it seems deatils mentioned in manuscript but not in abstract
- Introduction
- The introduction provides a solid rationale for the study; however, write the critical gaps and the objectives of this review need to be mentioned.
- Line Nos. 55, 56 and 57: The phrasing in the abstract lacks clarity, particularly regarding the relationship between antibody-mediated responses, challenges, and emerging diseases. The sentences appear ambiguous, and the linkage between these concepts needs to be articulated more precisely.
- Comment on Figure 1: Figure 1 appears incomplete, as it primarily highlights selected aspects without incorporating other major classes of pathogens. Since the manuscript discusses antibodies in the broader context of infectious diseases, the figure would be more comprehensive and informative if it also included bacterial, fungal, and parasitic pathogens in addition to viral agents. Expanding the figure to represent the diverse spectrum of host–pathogen interactions would cover wide area of pathogen.
- Line Nos. 33-34: The sentence presented in lines 33 and 34 could be reformulated for greater clarity and precision. The current wording may cause ambiguity.
- The inclusion of more figures and tables will improve the clarity and accessibility of the manuscript. As complex processes such as antibody generation, signaling pathways, and effector functions are more effectively conveyed through schematics, flowcharts, and tabular summaries. Such additions would enhance readability and provide readers with a more concise and integrated understanding of the subject matter.
- Materials and Methods
- The authors need to mention in Materials and Methods what is the selection criteria followed for inclusion and exclusion of articles in this review. Further, the authors need to mention what is the time period followed for inclusion of articles in this review. The authors should justify, why the authors have selected only PubMed articles because many scientific databases are available for retrieving the scientific articles. Further, all the article are not indexed in PubMed.
- Role of antibodies in emerging and re-emerging infections
- In Roles of antibodies section classified as protective and pathogenic. The Pathogenic role may cause misleading phrase; alternate subheading may be more informative and suitable.
- As majorly classified in two broad categories a) Protective and Pathogenic
- A tabular and diagrammatic presentation role and signaling pathway is encouraged to incorporation.
- Protective category: Antibody generation and functions are presented well but a more detail on B-cell activation, germinal center dynamics, class-switch recombination, somatic hypermutation, and Fc-mediated effector with its recent advances, signaling pathways, and pathogen-specific variations needs to incorporate in manuscript.
- All subheadings in section 2.1 primarily focus on antibody responses to viral infections. To maintain balance and breadth, it would be valuable to discuss antibody responses against other pathogenic organisms, including bacteria, fungi, and parasites. Alternatively, if the intention is to emphasize viral infections, the section should be restructured accordingly and complemented with a detailed discussion on the roles of cytokines and interferons in shaping the antiviral immune response.
- Line No 128-129
- Check for repetition of reference (17) in same para
- Section 2.2: Pathogenic Autoantibodies
- Include recent advancement in engineered antibodies and vaccine with respect to antibody response in terms of pros and cons which will support section 2.2
- Line No 189-193 and subsequent lines: The manuscript addresses autoimmune diseases, it would be important to provide a detailed description of the relevant autoantibodies. Autoantibodies are central to the pathogenesis of many autoimmune disorders, as they can directly mediate tissue damage or form immune complexes that drive inflammation. In addition, they serve as valuable diagnostic and prognostic biomarkers and are often incorporated into clinical classification criteria (e.g., anti-dsDNA in systemic lupus erythematosus, anti-CCP in rheumatoid arthritis, anti-TPO in autoimmune thyroiditis). Including a focused discussion or summary table of disease-specific autoantibodies would significantly enhance the scientific depth and clinical relevance of the manuscript.
- Line no 257-265: Correct the grammatical mistakes and rewrite the para
- In section 4: Factors Influencing Antibody-Mediated Immunity
- Incorporate all factors in terms of pathogens (Bacteria, Virus, Fungi..etc) which results in vaccine insufficiency e.g Tuberculosis.
- As this is a review article, the discussion on antibody generation and functions are presented well but a more detail on B-cell activation, germinal center dynamics, class-switch recombination, somatic hypermutation, and Fc-mediated effector with its recent advances, signaling pathways, and pathogen-specific variations incorporation would greatly enhance the scientific rigor and provide readers with a more comprehensive understanding of antibody biology in infectious diseases.
- Conclusion
The conclusion is not written in a holistic manner. There is noticeable repetition of representative terms, which should be avoided. Moreover, while the manuscript primarily addresses antibody responses in infectious diseases, the conclusion disproportionately emphasizes immunotherapy and vaccines. A more balanced conclusion that integrates antibody responses across all major classes of pathogenic organisms (viruses, bacteria, fungi, and parasites) would better reflect the overall scope of the manuscript.
11. References
Authors are requested to check and format once again all the references according to journal
format especially while abbreviating the Journal names. In some References, volume, issue
and page numbers are missing.
12. Other minor comments
The grammatical mistakes need to be avoided throughout the manuscript.
Comments on the Quality of English Language
The grammatical mistakes need to be avoided throughout the manuscript.
Reviewer 3 Report
Comments and Suggestions for Authors
This article has an interesting title “The Yin and Yang of antibodies in infectious diseases”. Indeed, the title is the best part of the paper, except that “infectious diseases” is too broad as the discussion is limited viral infections. The abstract is also good, minus the last two sentences which sound good but have little content.
The narrative is lengthy and contains unnecessary elements which make it difficult to follow the point. For one thing, there are many sentences which are in layman’s language and populistic, starting from the first one “In the face of emerging and re-emerging infectious diseases, the world is reminded of the critical importance of the humoral immune response”. The authors should critically search and delete such sentences which are more fit to news media or, otherwise, school textbooks. Indeed, the whole section of (1.) Introduction could well be deleted and the paper started with (2.) Role of antibodies in infections. There is no need to mention “emerging and re-emerging”, which sounds populistic.
The second type of element which could be deleted or shortened is textbook like references, such as the list of complement components or historical references to rheumatoid factor.
The section (5.) Conclusions is confusing as it is not well derived from the main text. This part should be reconsidered and totally rewritten.
The section (6.) Limitations is not really necessary. There is no need to make excuses for citing animal studies. It is best to delete the whole section.
Round 2
Reviewer 2 Report
Comments and Suggestions for Authors
I have carefully reviewed the manuscript along with the authors’ responses. Mos of the reviewer comments have been thoroughly addressed, and the authors have made thoughtful and meticulous revisions to the manuscript.
However, few queries remain unaddressed while revising the manuscript in haste.
The authors need to mention in separate heading like Materials and Methods what is the selection criteria followed for inclusion and exclusion of articles in this review. Further, the authors need to mention what is the time period followed for inclusion of articles in this review.
Author Response
- Summary
Thank you very much for taking the time to review this manuscript. Please find the detailed responses below and the corresponding revisions/corrections in highlighted in the re-submitted file.
- Point-by-point response to Comments and Suggestions for Authors
Comment: The authors need to mention in separate heading like Materials and Methods what is the selection criteria followed for inclusion and exclusion of articles in this review. Further, the authors need to mention what is the time period followed for inclusion of articles in this review.
Response: Thank you for your feedback. In response to your comment, we added the "Materials and Methods" section including the selection criteria and the time period for the inclusion of articles.
Added Text (lines 85-97): 2. Materials and Methods
This manuscript is a narrative and descriptive literature review designed to synthesize the dualistic 'Yin-Yang' roles of antibodies in viral infectious diseases. A structured, conceptually driven literature search was conducted across PubMed, Scopus, and Google Scholar using keywords related to antibody functions. The search prioritized publications from January 2015 to present to ensure currency, while also including seminal foundational articles for historical context. Selection was limited to peer-reviewed original research and authoritative reviews in English, excluding articles with poorly described methodologies. The selected evidence was then thematically synthesized, organizing findings to first establish the protective 'Yin' mechanisms (e.g., neutralization, ADCC) and subsequently explore the detrimental 'Yang' roles (e.g., ADE, autoimmunity), thereby constructing a coherent narrative that connects fundamental immunology to clinical implications and the design of future interventions.

Reviewer 3 Report
Comments and Suggestions for Authors
the revised version is much improved and acceptable
Author Response
Thank you for your time and for reviewing our manuscript. We are delighted that you found the revised version to be much improved and acceptable. We appreciate your valuable contributions, which have helped us strengthen the paper.
